# Digital (R)Evolution: Open-Source Softwares for Orthodontics

**Fabio Federici Canova** [1,*], **Giorgio Oliva** [2], **Matteo Beretta** [1] and **Domenico Dalessandri** [2]

1   Department of Medical and Surgical Specialties, Postgraduate Orthodontic School, Radiological Sciences and Public Health, University of Brescia, Piazzale Spedali Civili 1, 25123 Brescia, Italy; teoberet@libero.it

2   Department of Medical and Surgical Specialties, School of Dentistry, Radiological Sciences and Public Health, University of Brescia, Piazzale Spedali Civili 1, 25123 Brescia, Italy; giorgio.oliva1@gmail.com (G.O.); domenico.dalessandri@unibs.it (D.D.)

*   Correspondence: fabiofedericicanova@gmail.com

**Abstract:** Among the innovations that have changed modern orthodontics, the introduction of new digital technologies in daily clinical practice has had a major impact, in particular the use of 3D models of dental arches. The possibility for direct 3D capture of arches using intraoral scanners has brought many clinicians closer to the digital world. The digital revolution of orthodontic practice requires both hardware components and dedicated software for the analysis of STL models and all other files generated by the digital workflow. However, there are some negative aspects, including the need for the clinician and technicians to learn how to use new software. In this context, we can distinguish two main software types: dedicated software (i.e., developed by orthodontic companies) and open-source software. Dedicated software tend to have a much more user-friendly interface, and be easier to use and more intuitive, due to being designed and developed for a non-expert user, but very high rental or purchase costs are an issue. Therefore, younger clinicians with more extensive digital skills have begun to look with increasing interest at open-source software. The aim of the present study was to present and discuss some of the best-known open-source software for analysis of 3D models and the creation of orthodontic devices: Blue Sky Plan, MeshMixer, ViewBox, and Blender.

**Keywords:** digital orthodontic; open source software; Blue Sky Plan; Meshmixer; Blender; ViewBox

## 1. Introduction

Among the innovations that have changed modern orthodontics, the introduction of new digital technologies in daily clinical practice has had a major impact, in particular the use of 3D models of dental arches. The possibility for direct 3D capture of arches using intraoral scanners or the digitisation of plaster models has brought many clinicians closer to the digital world. The advantages of this revolution are many: the 3D models can be used for dental analysis as precisely as stone cast [1,2], and virtual models simplify storage procedures by eliminating stone cast.

Digital 3D models are generally saved and displayed in the STL format (Standard Triangle Language), which has now become the standard format for model evaluation. The STL format does not have color maps, so it is monochrome. If you want to include the color texture as well, the reference formats are obj (Object File Wavefront 3D) and ply (Polygon File Format).

The use of digital models has made it possible to modify the diagnostic approach, improving and simplifying classic orthodontic analyses [3–6] and allowing virtual simulation of the patient and dental movements [7,8]. In addition, the decreased invasiveness of the method and the possibility of overlapping the models allows greater control of therapeutic progress [6,9,10]. The possibility of using 3D printing machines and very high precision resins allows the creation of any orthodontic products [11]. There is also the possibility of managing a full digital workflow: there are many devices that can be made without going

through the prototyping of a model, but rather by designing the entire device digitally with dedicated software [12–14]. Finally, the use of digital models also plays a fundamental role in scientific research [15–17], since stereolithography (STL) surfaces can be compared and analysed without loss of detail by different research groups in different places.

This digital revolution has clearly affected the orthodontic world, but there are some negative aspects, including the need for the clinician to learn how to use new software. In this context, we can distinguish two main software types: dedicated software (i.e., developed by orthodontic companies) and open-source software. Dedicated software tends to have a much more user-friendly interface, and be easier to use and more intuitive, due to being designed and developed for a non-expert user, but very high rental or purchase costs are an issue. Therefore, younger clinicians with more extensive digital skills have begun to look with increasing interest at open-source software. These programs represent a very interesting resource: some software is totally free or marketed on a 'pay-as-you-go' basis. However, disadvantages include a fairly long learning curve.

The aim of this study was to present and discuss some of the best-known open-source software for analysis of 3D models and the creation of orthodontic devices: Blue Sky Plan, MeshMixer, ViewBox, and Blender (Table 1).

**Table 1.** Summarising table.

| | Blue Sky Bio | Meshmixer | Viewbox | Blender |
|---|---|---|---|---|
| **download at** | www.blueskybio.com | www.meshmixer.com | www.dhal.com/viewbox.htm | www.blender.org |
| **OS operating system** | macOS/Windows | macOS/Windows | Windows | macOS/Windows |
| **cost** | free to use, only some export on local system are charged | completely free | waiting time of 30 sec on free version to use some tools | blender is free, the add-on blenderfordental is charged |
| **main use** | surgical guide for palatal TADs, analyse models, orthodontic set-up | repair and edit mesh | analyse 3D models, repair and edit mesh | edit 3d mesh, analyse models |

## 2. Blue Sky Plan

Blue Sky Plan (BSP) is an advanced treatment planning software distributed by BlueSkyBio (www.blueskybio.com, Blue Sky Bio, LLC—800 Liberty Drive, Libertyville, IL, USA, accessed on 22 May 2021), which was developed primarily for guided implant surgery. At the time of writing, the 4.8.2 release is available for 64-bit PC platforms, while it is still possible to use the software for the iOS platform, but it is no longer released for MacOS. It is a 'pay-per-use' software: it can be downloaded and used free of charge, without limitations and without license renewals. All software updates are free and available in many languages.

Inside the software there are several modules, extremely useful for both orthodontic and prosthetic fields [18]. Most of the functions are free, and only the local saving of some STL files, such as surgical guides, is paid for at a reduced cost. Among the modules in the program, the most significant are the viewer module, the orthodontic module, and the guided surgery module.

### 2.1. Viewer Module

Blue Sky Plan allows the user to import Digital Imaging and Communication in Medicine (DICOM) file sets for viewing in multi-window mode. The most common views are available (axial, coronal, implantology cuts) and easily switchable, with key combinations. Furthermore, 3D rendering is performed well since some presets are available, and this makes visualisation very efficient. Furthermore, automatic matching between the STL surface and the DICOM set allows to obtain a link between the volumes in a very short time. In the 'model modification' section, it is possible to edit the models of the patient's arches, for example by orienting the models in the spatial plane, to create articulation pins

that can be printed, or even to match two different STL surfaces, for example to evaluate the modification of arch shape (Figure 1A,B).

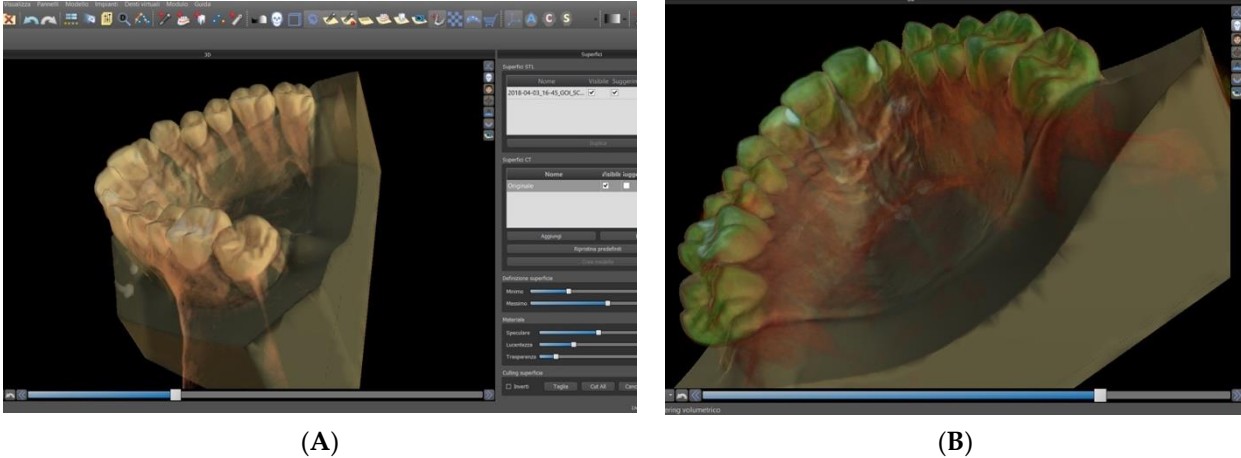

(**A**)       (**B**)

**Figure 1.** (**A**,**B**) STL and DICOM viewer.

### 2.2. Orthodontic Module

The orthodontic module allows the planning of digital set-up of the arches, which can also be used for 'in-office' production of aligners (Figure 2A,B). The orthodontic module allows the user to do the following:

- Import an STL file and segment the dental elements.
- Carry out alignment of the arches.
- Import the antagonist arch and view the contact points using colorimetric maps.
- Perform inter-proximal-reduction (IPR).
- Import a CBCT and superimpose it on the scanned arch.
- Check all alignment steps and save the STL file for clear aligner therapy.

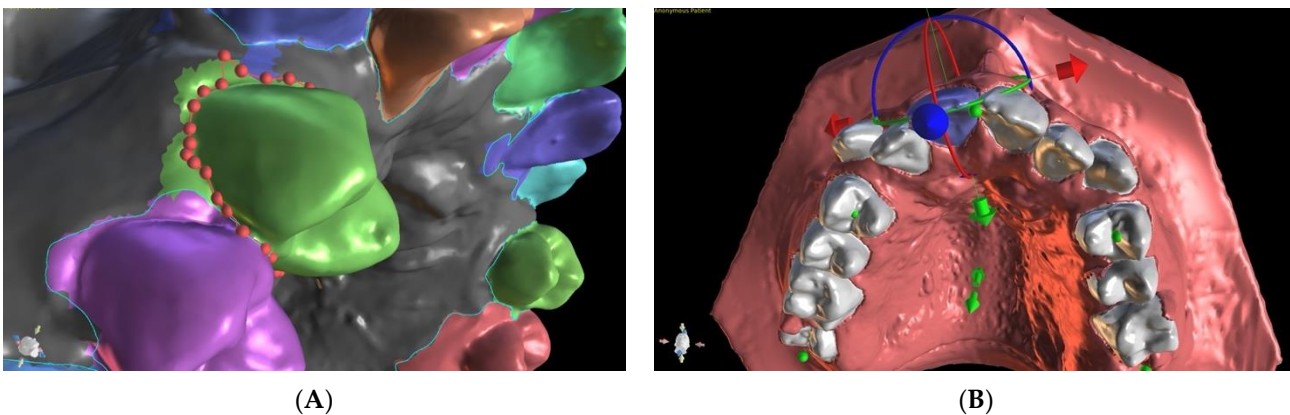

(**A**)       (**B**)

**Figure 2.** (**A**,**B**) In the orthodontic module there are the teeth segmentation functions and the set-up commands.

The segmentation of teeth is an extremely fast and intuitive process, thanks also to the navigation menu that indicates the steps to be carried out: for each element, only the mesial and distal points must be marked, then the software will automatically calculate the segmentation curve using the contour gingival as a reference. However, the user is asked to correct the curve if there are smudges. Once the teeth have been segmented, it is possible to carry out the virtual set-up using a series of very interesting features:

- The ability to view and control stripping.

- Perform tooth movement functions both graphically with the handles, and with keyboard entry for greater precision.
- The ability to view contacts with the antagonist arch.
- The possibility to save and print the various movement steps in STL format for the production of transparent aligners. This feature must currently be paid for.

### 2.3. Guided Surgery Module

This is the main software module, and it allows the user to design implant surgical guides using a few simple steps. Although the module was created for implant-prosthetic purposes, in orthodontics it can be effectively used for the design of templates for palatal or vestibular temporary anchorage devices (TADs) for skeletal anchorage, thanks to the fact that it is possible to absolutely customise the size of the implant to be inserted. It starts with the import of two objects: a model scan in STL format, and DICOM files obtained from the CBCT exam. The software will then automatically match the two surfaces.

Next, the TADs are inserted virtually, not using the available libraries, but by editing a new TAD that will be customised according to systematic choice. In this way, by modifying the apical diameter, cervical diameter, and collar height data, it is possible to use BSP with any implant company.

The entire design of the surgical guide is very easy, intuitive, and free. Only the export of the guide in STL format is subject to a fee at a very low cost. Finally, through the functions of the GSR model, it is possible to generate a perforated model through the automatic Boolean subtraction of the STL volume of the TADs or the analog. In this way, BSP can be effectively used not only for printing the guide, but also for printing a perforated model to be used for the 'one-step' insertion of orthodontic devices (Figure 3A–C).

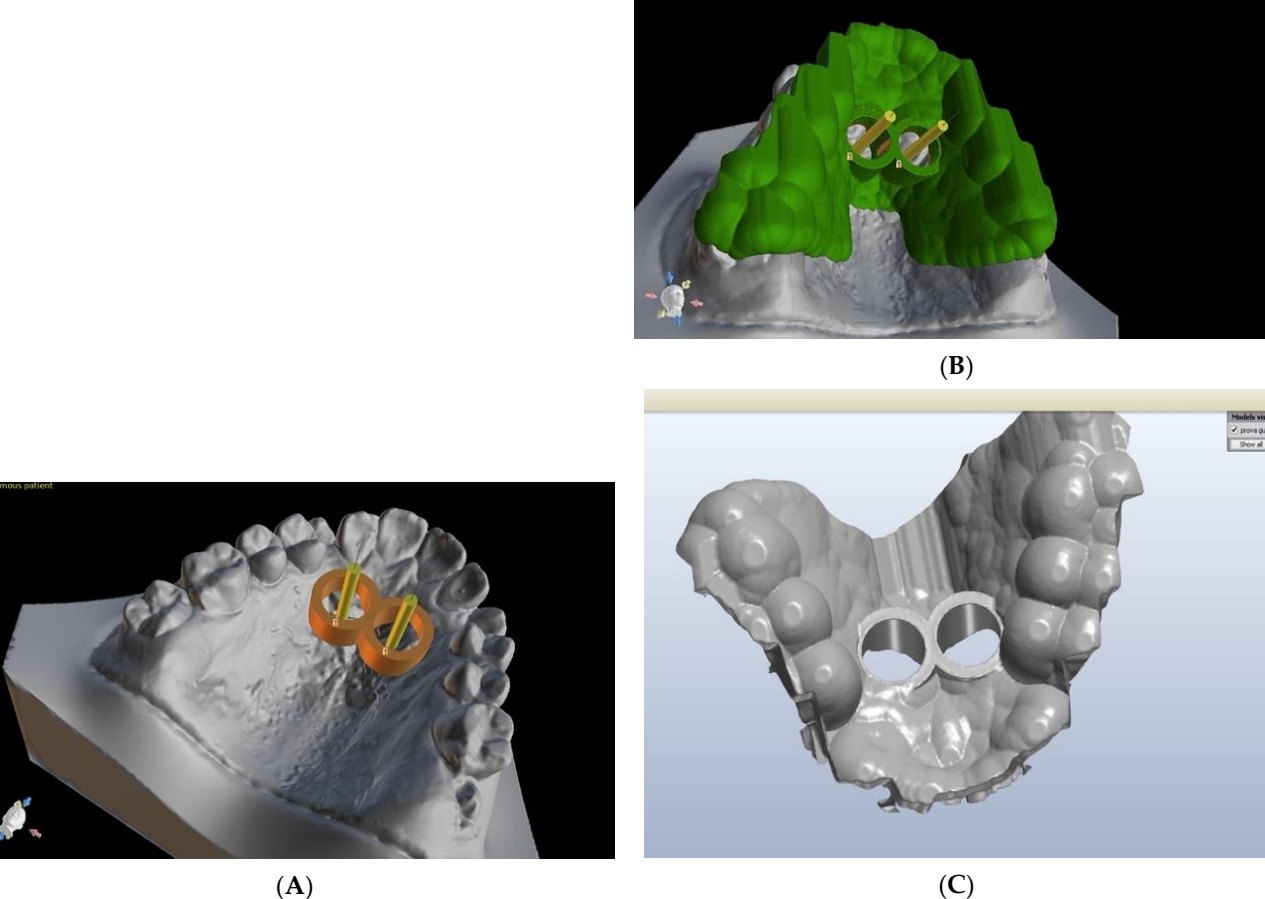

**Figure 3.** (**A**–**C**). Surgical module for TADs guided insertion.

### 3. Meshmixer

Autodesk MeshMixer (Autodesk, Inc., Mill Valley, CA, USA) is an open-source software to create and edit 3D objects with a simple and intuitive interface. It is one of the many useful programs produced by Autodesk, a leading software house in the 3D design sector. At the time of writing, it is available as version 3.5, downloadable from the website www.meshmixer.com (accessed on 22 May 2021), and is available for both Windows PC and MacOS, completely free of charge and with a detailed user manual. The name is self-explanatory: the program allows the mixing of several meshes together in order to obtain new objects with a characteristic design. It also allows the user to 'sculpt' our 3D object using a series of tools such as volume brushes, surface brushes, symmetry tools and others. Since the application is free and non-commercial, its interface is perfectly designed to be easy to use. For those who want to use MeshMixer more seriously, many tutorials are available on the official website and on YouTube.

In the orthodontic field, MeshMixer has multiple applications due to its versatility and the absence of registrations and costs [19–21]. MeshMixer allows the user to:

- Repair holey meshes by closing holes in the STL file.
- Edit STL models when there are surface aberrations linked to the incorrect reconstruction of portions of the tooth (for example in the case of metal or ceramic teeth that reflect the scanner light in a non-perfect way, and consequently the reconstruction software can interpolate an incorrect surface).
- Convert models from open shell to closed surface format.
- Mount the two arches in the virtual articulator in order to simulate and evaluate the patient's occlusion, as part of a virtual mock-up.
- Simulate the surgical set-up, moving an arch with respect to the antagonist in an extremely simple way.
- Design orthodontic bands that can then be printed with the laser-melting technique [22].
- Design any other prosthetic or orthodontic structure (lingual retainers, retaining devices, palatal bars, etc.), which can be saved in STL format and printed in the most appropriate material.
- Create a virtual mock-up of the patient using the libraries with ideal tooth shapes made available for free. Drawing, saving the project, and exporting the STL file are totally free.

Even with these advantages, and despite the relative ease of use for a user accustomed to CAD/CAM design, MeshMixer may not be easy to use for an orthodontist or dental technician with low-level computer skills (Figure 4).

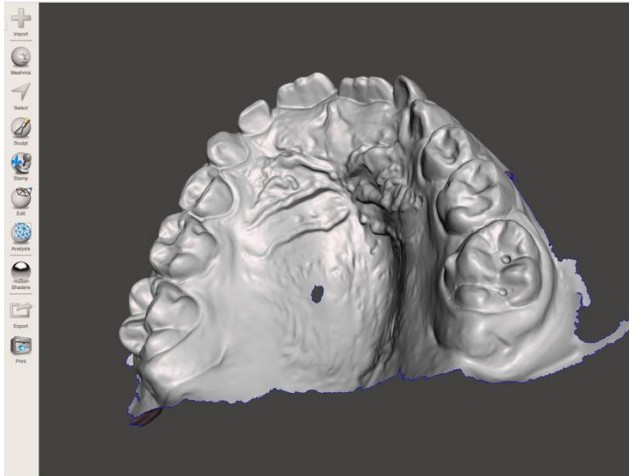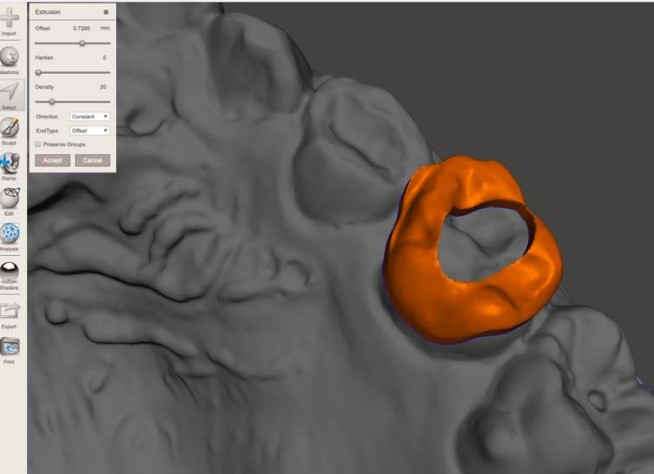

**Figure 4.** Meshmixer can be used to repair meshes or create custom bands.

## 4. Viewbox

ViewBox is a software developed by DHAL (dHAL Software, Kifissia, Greece), available in the 4.1.0.12 release for the Windows platform only at the time of writing, and downloadable from the website www.dhal.com/viewbox.htm (accessed on 22 May 2021). In the free version, all the native tools of the program are useable, but there is a waiting time of 30 s to use some tools. Therefore, the free version can be effectively used without any limitations for model analysis (Figure 5).

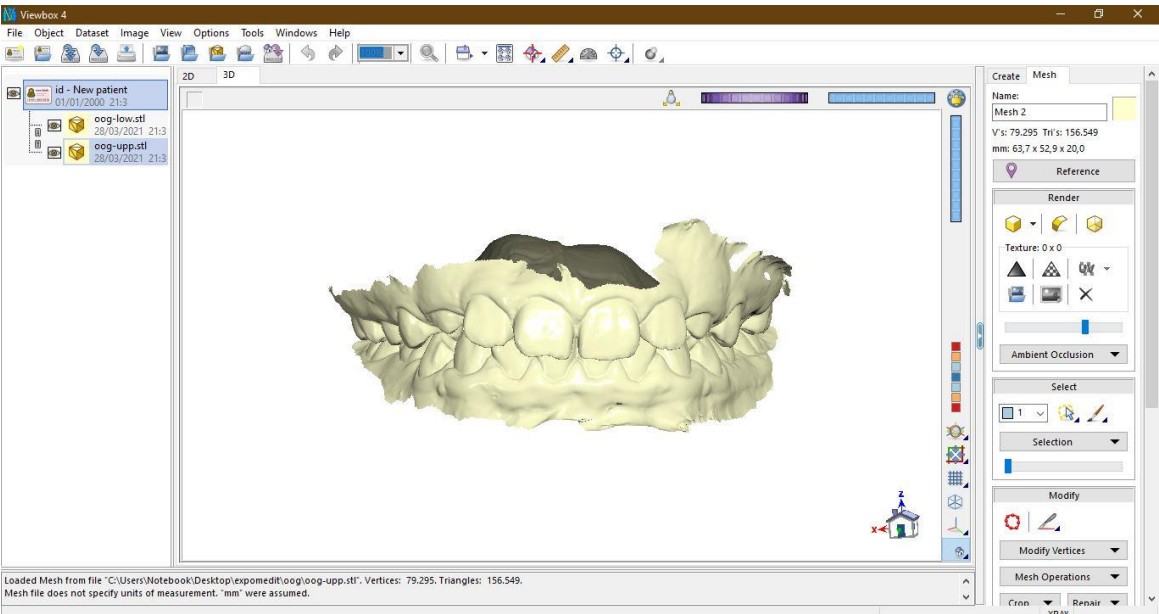

**Figure 5.** Models opened in Viewbox.

In the latest version of the software, many specific functions are included for 3D models. To these are associated all analyses used for 2D images such as teleradiographies or simple photographs. The software allows the user to repair, modify, or improve the mesh of the model, and prepare the model for printing (Figure 6).

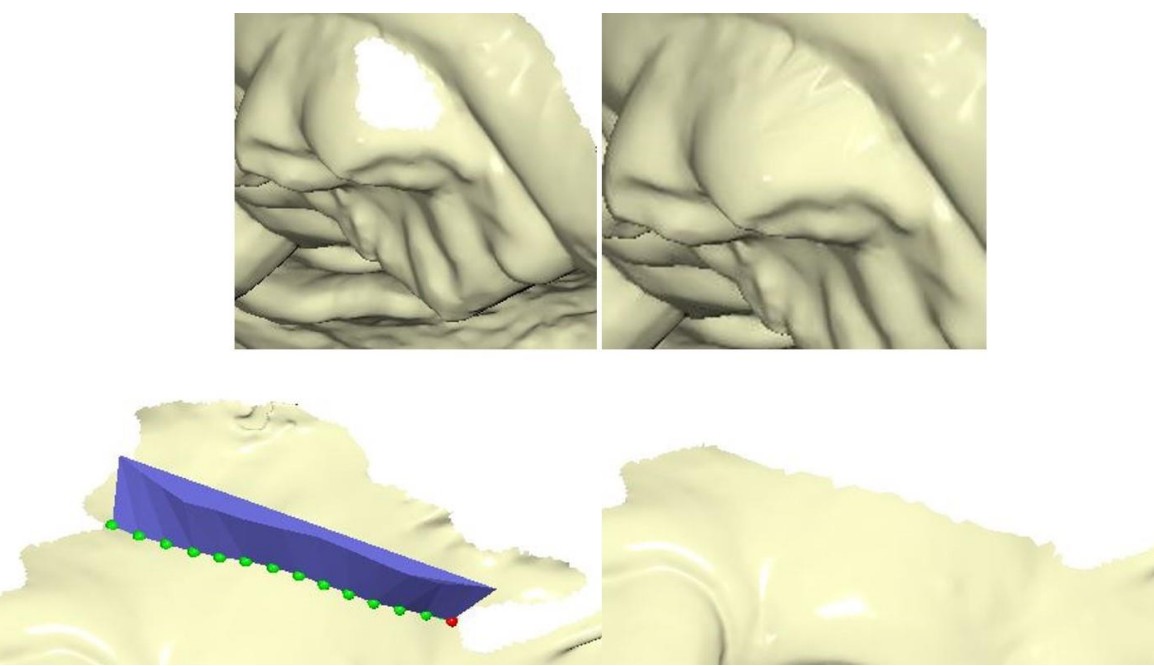

**Figure 6.** Mesh repair and scan edge optimisation.

ViewBox is able to import and analyse 3D models from different formats (STL, OBJ, PLY). The main advantage of ViewBox is the possibility to store all patient data in a single database, whether photos, X-rays, or scans. In the daily routine, this allows for greater ordering of patient data archiving and greater ease of information backup. The software is characterised by an impressive versatility regarding the range of analyses possible (Figure 7).

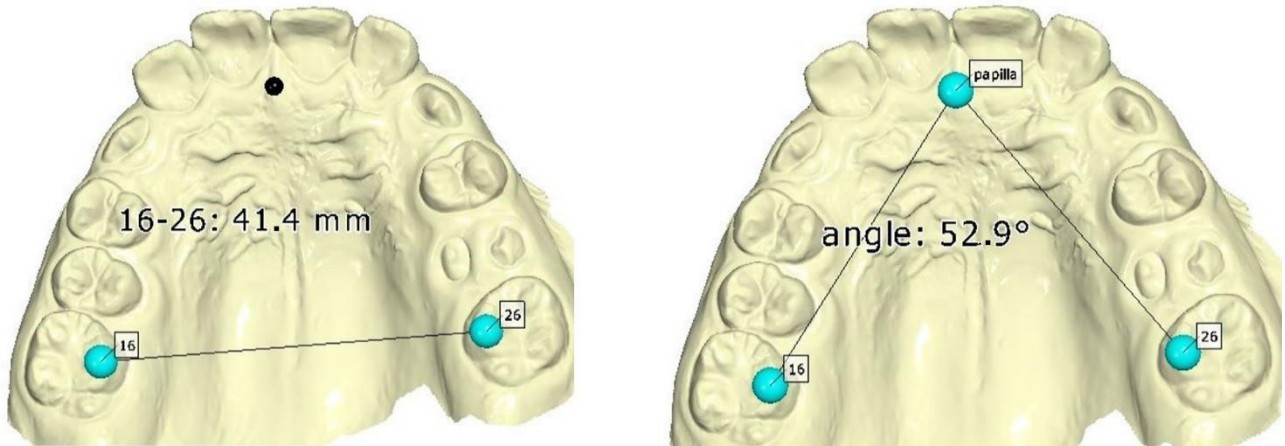

**Figure 7.** Examples of linear and angular measurements.

Generating a customised analysis is very simple, and numerous tutorials can be found on the software website. Measurements can be linear and angular, and involve points selected manually by the operator or points built by software. In addition to performing simple and rapid analyses, ViewBox allows the development of summary reports of analyses. It is possible to set reference values in order to more easily analyse the results obtained. In addition to classic metric analyses, it is possible to make superimpositions of models in order to analyse the dental movements occurring during therapy. Unlike most software, tooth displacements can be visualised tooth-by-tooth by carefully observing the individual position variations (Figure 8).

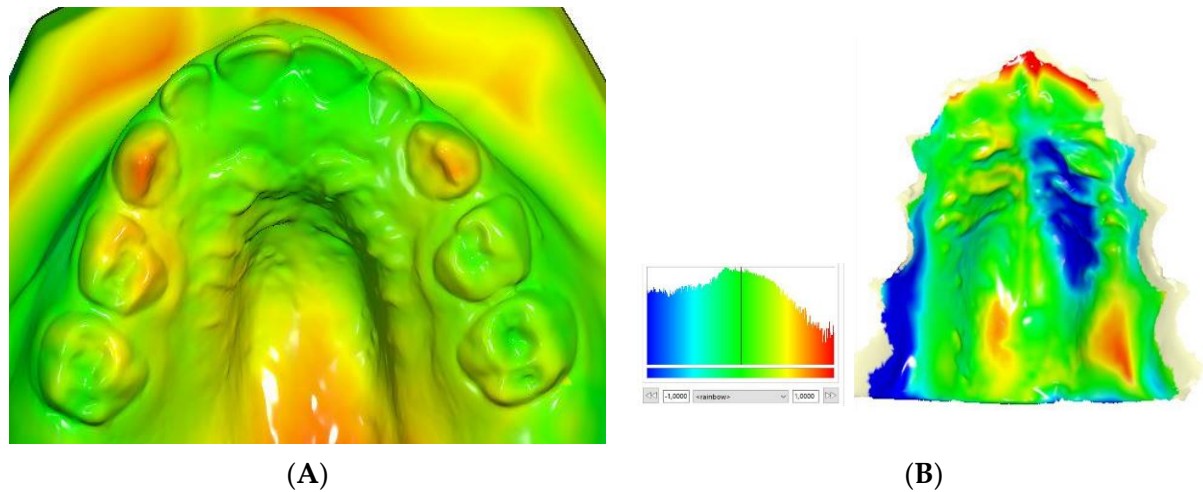

**(A)**         **(B)**

**Figure 8.** (**A**,**B**) Example of a colormap created by superimposing two models.

In addition to creating individual patient reports, it is possible to export directly to spreadsheets analyses carried out on different patients or on different models of the same patient. All analyses and patient data are saved in an archive file that can be easily copied and saved on an external disk to protect against data loss.

## 5. Blender and the Blenderfordental Add-On

Blender (Blender Foundation, Amsterdam, The Netherlands) is one of the most widely used programs for processing and editing 3D images [19,23]. It is available at www.blender.org (accessed on 22 May 2021) for Windows, iOS, and Linux platforms. The software is completely free and open source, but compared to previous programs, it is much more complex to use in its original version. It was developed for pure 3D drawing and requires CAD/CAM modelling skills that are not common among orthodontists. To overcome this complexity, Blenderfordental (www.blenderfordental.com, accessed on 22 May 2021) has developed add-ons that transform Blender into a platform that is very easy to use and more complete. The Blenderfordental basic module (model designer) is the first module to obtain if you want to use the software, and it has been specially designed for the dental world. With this module it is possible to carry out many useful operations for both the analysis of cases and the modification of models, and for preparation for 3D printing (Figure 9).

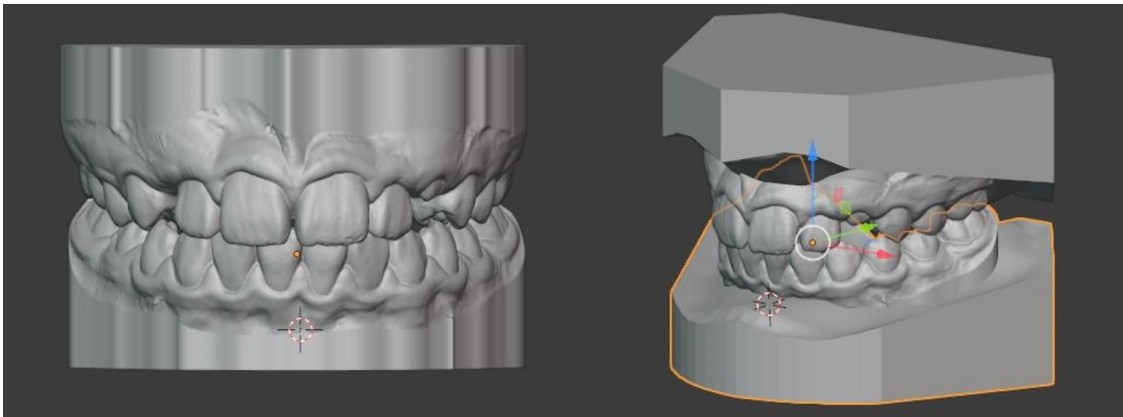

**Figure 9.** Ready-to-print molds with plain bases or tweed bases. The models can be emptied or connected by means of a vertical articulator.

Models can be loaded and oriented in almost any format currently used for 3D files. In the model designer, orientation is performed by the operator selecting four points, or by adding the articulator module, it is possible to obtain a more complex model alignment thanks to the use of patient photos and extraoral references. Once oriented correctly, the model can be cleaned of imperfections thanks to the vast library of brushes present in blender, similar to those offered by MeshMixer, or by using the hotkeys provided by Blenderfordental. It is therefore simple to inspect the mesh, close any holes present on it, eliminate scan defects, and create clear and sharp edges. It is also possible to create bases for the digital plinth of 3D models, and to prepare the model for printing. Once the bases have been created, the software allows the user to perform many operations on the models, including linear, angular and arch measurements, creating removable dental elements, separating individual teeth, analysing occlusal contacts, analysing lateral and protrusive movements, and simulating the advancement of the mandibular arch with a virtual articulator (Figures 10–13).

The software can be further integrated with the purchase of other useful modules, for example to reconstruct occlusal tables, to create templates for mock-up or design of occlusal splints, or removing the brakes. There are advanced functions for inter-model study: different models can be aligned with each other, the relative movements that have occurred can be analysed, and variation in arch shape can be evaluated. It is also possible to create guides for implant insertion with functions similar to those already described for BSP. To prepare models for printing, the software allows you to empty them while maintaining the desired thickness, to examine the direction of the normals, and to inspect the mesh for any distortions and sharp corners and for export. Finally, using only the

basic module, it is possible to add articular pins to models that allow bringing the printed models back into the correct and desired occlusion.

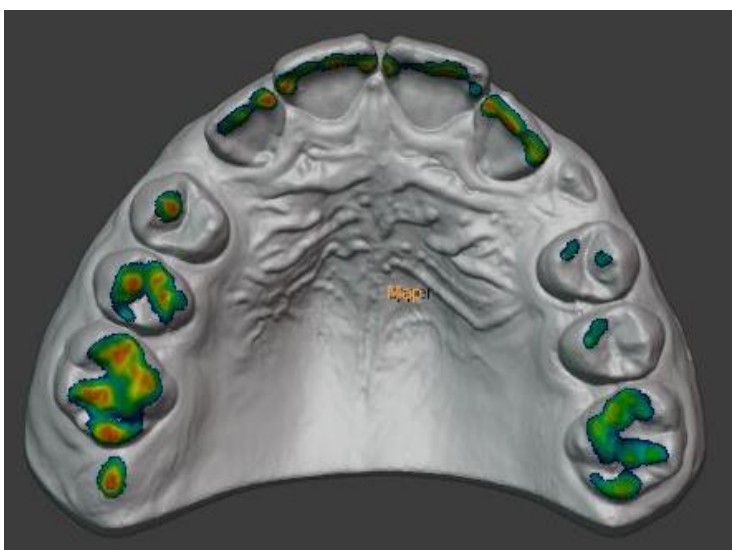

**Figure 10.** Occlusal contacts analysed by the software.

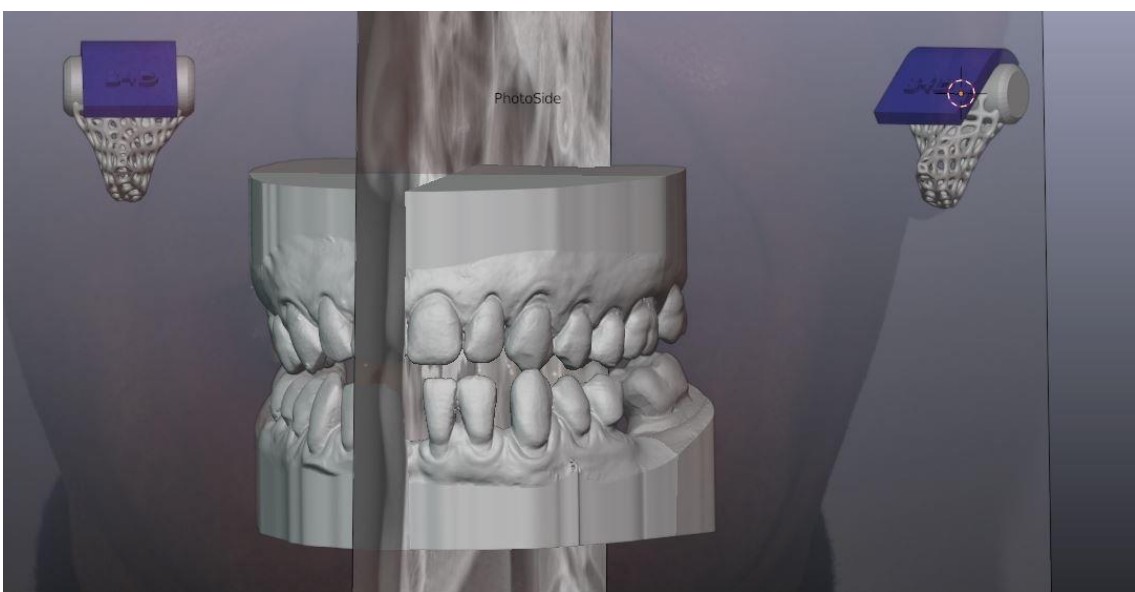

**Figure 11.** Virtual articulator.

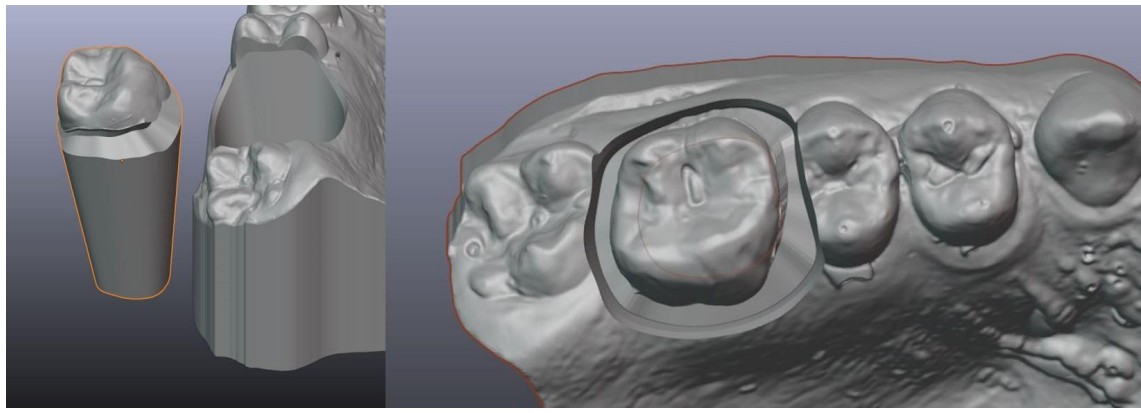

**Figure 12.** Templates prepared for inserting bands.

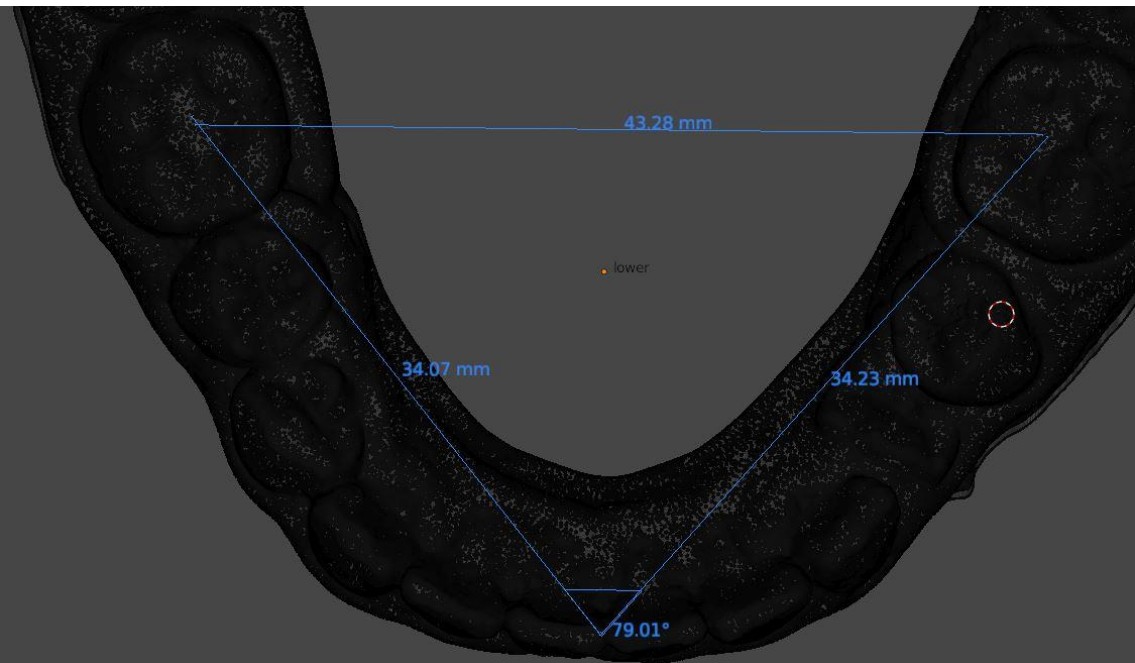

**Figure 13.** Examples of linear and angular measurements.

## 6. Conclusions

The digital revolution of orthodontic practice requires both hardware components and dedicated software for the analysis of STL models and all other files generated by the digital workflow. Some are closed-source software, developed by the scanner manufacturers themselves, while others are open-source software.

To use open-source software effectively, it is important to be able to switch from one program to another, and to use several programs at the same time, since each has specific characteristics and functions that must be integrated.

While low operating costs are the main advantage, open-source software is generally associated with a longer learning curve, since programs are not dedicated to particular instruments. However, free online tutorials are available that allow the user to quickly learn the functions and commands of programs, and thereby automate many processes and save time. The communities and forums that are created around open-source software also help in its development and dissemination.

New open-source programs will lead to new perspectives in the dental industry. Indeed, if the costs for accessing modern digital technologies remain high, even though they are likely to decrease in the coming years, the use of open-source software can help to bring new generations of dentists closer to the digital world.

**Author Contributions:** Conceptualization, F.F.C. and G.O.; methodology, M.B.; software, F.F.C. and G.O.; validation, D.D. and M.B.; formal analysis, D.D.; investigation, F.F.C.; data curation, G.O.; writing—original draft preparation, F.F.C. and G.O.; writing—review and editing, M.B. and D.D.; supervision, D.D.; project administration, M.B. All authors have read and agreed to the published version of the manuscript.

**Funding:** This research received no external funding.

**Institutional Review Board Statement:** Not applicable.

**Informed Consent Statement:** Not applicable.

**Conflicts of Interest:** The authors declare no conflict of interest.

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
