# Peer review of "Digital (R)Evolution: Open-Source Softwares for Orthodontics"

_applsci, doi:10.3390/app11136033_

Round 1

Reviewer 1 Report

The topic is interesting and really up-to-date. In my opinion a lot of orthodontic practitioners will find ot useful. The paper is well organized including the following structure: abstract, introduction, main body (divided into sections describing particular types of software) and conclusions. The number of references is relevant to the subject of research.

I suggest publication after the authors have considered the comments above and the following minor remarks:

Line 13 – space before the word “however’

Line 30- should be [1,2] instead of [1], [2]

Line 31 – can you explain what the word “precise” mean? Are they more accurate than alginate impressions? How about A-silikone impressions? It concers the precision of scans obtained directly from oral cavity or either the scans of plaster models?

Line 55 I recommend to change the word “present” because of repetition

Line 200 sequence of citation – should be [19,23]

In my opinion if the authors include the summarizing table with a comparison of presented software with the reference to their orthodontic implementation, the reader will find easer helpful information.

Author Response

Dear colleague, thank you for reviewing my manuscript.

I will correct the text according to your instructions as soon as possible.

kind regards,

Fabio Federici Canova

Reviewer 2 Report

Dear Authors,

 this manuscript aims to summarize and discuss about open sources software for 3D models and orthodontics devices. A very interesting and original topic.

I have few comments for you

Line 32-33 please add a paragraph on which you explain how the 3D model is created (Standard Triangle Language) STL files.

Line 72 blue sky plan add city and nation

Line 80-81 some figures could facilitate the explanations

Line124 Meshmixer add city and nation

E.G. (Worldwide Headquarters. Autodesk, Inc. 111 McInnis Parkway San Rafael, CA 94903. USA)

Line 161 DHAL please add (HAL Software, 6 Menandrou Street, Kifissia 14561, Greece.)

Line 160 ViewBox please add city and nation

Line 192 Figure 8 please add a measurement scale e.g., mm

Line 198 as the previous Blender city and nation

The conclusions are in line with the topic.

kind regards 

Author Response

(The authors gave the same response as above.)
